# Strength and Microstructure of Geopolymer Based on Fly Ash and Metakaolin

**DOI:** 10.3390/ma15103732

**Published:** 2022-05-23

**Authors:** Salim Barbhuiya, Edmund Pang

**Affiliations:** 1Department of Engineering and Construction, University of East London, London E16 2RD, UK; 2School of Civil and Mechanical Engineering, Curtin University Australia, Perth 6845, Australia; edmund.k.pang@student.curtin.edu.au

**Keywords:** geopolymer, fly ash, metakaolin, microstructure, compressive strength

## Abstract

The production of Portland cement is widely regarded as a major source of greenhouse gas emissions. This contributes to 6–7% of total CO_2_ emissions, according to the International Energy Agency. As a result, several efforts have been made in recent decades to limit or eliminate the usage of Portland cement in concrete. Geopolymer has garnered a lot of attention among the numerous alternatives due to its early compressive strength, low permeability, high chemical resistance, and great fire-resistant behaviour. This study looks at the strength and microstructure of geopolymer based on fly ash and a combination of metakaolin and fly ash. Compressive strengths were measured at 7, 14, and 28 days, and microstructure was examined using SEM and XRD.

## 1. Introduction

Researchers have been examining the impact of greenhouse gases on global warming for the past three decades. Rising sea levels, changes in ocean water that threaten marine life, shifting weather patterns, and ecological degradation are all results of global warming [1]. In the next 100 years, the global temperature is expected to rise by 3 °C, with a possible increase of 4.6 °C. As a result of these temperature increases, the sea level is anticipated to rise by up to 28 cm [2,3]. The International Panel on Climate Change (IPCC) believes that human activity is the most likely cause of observed warming since the mid-twentieth century. Given that people are a major cause of global warming, businesses such as coal power and the cement industry must reduce emissions.

A number of programmes and policies have been established in an attempt to minimise global carbon emissions and, as a result, global warming. The Kyoto Protocol [4] is an international agreement established by the United Nations Framework Convention on Climate Change (UNFCCC) in 1997 that commits signatories to emission reduction goals. Emissions trading is a concept for bolstering the Kyoto Protocol by giving economic incentives to companies, notably the concrete industry, to reduce carbon emissions [5]. The cost of carbon is expected to be around US $15 per tonne [6]. Portland cement production is widely acknowledged as a major source of greenhouse gas emissions [7,8,9,10,11,12]. This accounts for 6–7% of total CO_2_ emissions, according to the International Energy Agency (IEA) [13]. As a result, various initiatives have been made in recent decades to reduce or eliminate the use of Portland cement in concrete. Geopolymer has gotten a lot of attention among the many alternatives because of its early compressive strength, low permeability, high chemical resistance, and outstanding fire resistance behaviour [14,15,16,17,18,19].

The interaction of aluminosilicate material with alkaline solutions produces geopolymers. As a result, the two main components of geopolymers are source materials and alkaline liquids. Natural minerals such as kaolinite, clays, micas, andalousite, spinel, and by-products such as fly ash, silica fume, slag, rice husk ash, red mud, and so on might be used as source materials [20,21]. Fly ash and slag, in particular, have become popular source materials due to their high silica and alumina concentration and availability in landfill sites. A combination of fly ash and slag has also been employed by researchers.

During both the preparation and development stages, the metakaolin (MK)-based geopolymer has the advantage of being able to be manufactured consistently and with predictable qualities.

The plate-shaped particles, on the other hand, produce rheological challenges, increasing the processing complexity as well as the water need of the system. The geopolymer made from fly ash (FA) is often more durable and stronger [22]. FA-based geopolymers, on the other hand, are observed to have a substantial quantity of pores. The addition of metakaolin to FA-based geopolymers can result in a denser structure, which can result in increased strength. Because metakaolin alone produces a weak structure, it is combined with additional elements [23]. The strength, microstructure, and nanomechanical characteristics of several geopolymer mixtures based on metakaolin and fly ash are described in this work. At 7, 14, and 28 days, the compressive strengths were assessed. SEM and XRD were used to investigate the microstructure.

## 2. Experimental Programme

### 2.1. Materials, Mixing, and Curing of Samples

Based on earlier research [24], the mix design parameters of geopolymer mixes used in this investigation were chosen. The activating liquid was made up of a mixture of sodium silicate and sodium hydroxide solutions. The sodium silicate solution has a mass ratio of 2.61 (SiO_2_ = 30.0%, Na_2_O = 11.5%, and water = 58.5%) for SiO_2_ to Na_2_O. In a fume cupboard, sodium hydroxide pellets were dissolved in deionized water to make an 8 M solution.

The sodium silicate to sodium hydroxide ratios were varied between (1.0, 1.5, and 2.5). The activator solution was held at a constant mass ratio of 0.4 with the binder. One mix was made entirely of fly ash, while the other was made with 70% fly ash and 30% metakaolin. Both fly ash and metakaolin were obtained from local sources in Western Australia. The properties of fly ash and metakaolin are summarised in Table 1. The compressive strength testing was performed on 50 × 50 × 50 mm cube samples. After casting, the samples were demoulded and dried at room temperature of 20 (±2) °C with a relative humidity of 70 (±10) percent.

### 2.2. Test Methods

The compressive strength tests were done in line with ASTM C109 [25] Standard at 7, 14, and 28 days. A MIRA3 TESCAN scanning electron microscope (SEM) was used to investigate the morphology of the hardened samples). On a Siemens D500 BraggeBrentano diffractometer X-ray diffraction (XRD) studies were carried out across a 2θ range of 5–80. Using a Cu ka X-ray source, operating parameters for the XRD were set at 40 kV and 30 mA. The geopolymers’ crystalline phases were detected using a Powder Diffraction File (PDF).

## 3. Results and Discussion

### 3.1. Compressive Strength

Figure 1, Figure 2 and Figure 3 compare the compressive strength of FA and MK-based geopolymers and FA-based geopolymers at various curing ages and alkaline solution molar ratios. As can be shown, the majority of FA and MK-based geopolymer samples showed higher compressive strength than the FA-based geopolymers. MK has a stable chemical composition and particle size, which explains this. According to Parande et al. (2009) [26], unlike FA, MK is generated under regulated conditions, making it more reactive for geopolymerisation and hence improving geopolymer strength development. As a result, metakaolin’s tiny particle sizes allow for improved particle packing and can fill the holes left by unreacted FA.

When compared to the FA-based geopolymer, the FA and MK-based geopolymer samples demonstrated a maximum gain in strength of 30% when an alkaline solution molar ratio of 1.0 was utilised. The strength of the FA and MK-based geopolymers improved the greatest after 28 days, with the FA-based geopolymers improving by 9.8%. In Mix 3, the strength of both geopolymers was equal. The compressive strength difference between each increment in the alkaline molar ratio is noteworthy in this study. This shows that FA-based geopolymers are more affected by the alkaline solution ratio. In contrast, the FA and MK-based geopolymer performed better in the early stages. The strength of the FA and MK-based geopolymers, for example, was 13% higher at 7 days than the FA-based geopolymer. Zhang et al. (2014) [27] discovered that a 50/50 mixture of FA and MK geopolymers exhibited higher compressive strength after being heated than 100% FA geopolymers. The researchers also discovered that a geopolymer containing both FA and MK had a better compressive strength than geopolymers based on MK.

### 3.2. Microstructure

Figure 4 shows an example of several geopolymerisation phases. Mullite needles are found in the fly ash particle, indicating partial geopolymerization. The image also shows the remaining fly ash particles after the bond has been broken and geopolymer gels have formed and diffused throughout the structure, indicating that it has reached crystalline phase. Finally, the particle has a smooth surface, indicating that it did not participate in the geopolymerisation process and is, therefore, an unreacted FA particle. Numerous holes and compacted geopolymer gels may also be seen in the image.

Figure 5 and Figure 6 depict the microstructure of FA and MK-based geopolymers with 1.0 (Mix 1) and 2.5 (Mix 3) molar ratios of alkaline solution, respectively. The structure becomes less porous with fewer unreacted FA and MK particles as the alkaline solution molar ratio of the FA and MK-based geopolymer utilised in this study is increased, resulting in a stronger structure. The compressive strengths of Mix 1 and Mix 3 were 44 MPa and 62 MPa, respectively, as shown in Figure 1. By “extending the reaction”, an increase in the molar ratio of sodium silicate to sodium hydroxide, or an increase in the use of sodium silicate, aids in the geopolymerisation process. More FA and MK particles can react as a result, resulting in a more compact structure. Figure 5 also shows matrix porosity generated by FA or MK particles that failed to fully react during the geopolymerisation process.

Figure 7 and Figure 8 show the microstructure of FA-based geopolymers Mix 1 and Mix 3, respectively. Despite the fact that the compressive strengths of Mix 1 and Mix 3 are 35 MPa and 63 MPa, respectively, there are little to no differences between the two mixtures. The morphology of FA and MK-based geopolymers, on the other hand, differs significantly. Mix 1 of the FA-based geopolymer has more pores and unreacted pozzolan, which accounts for its lower compressive strength. Pores identify weak spots in the structure and generate routes for water to flow through. MK tends to fill pores when added to geopolymers because of its lower particle size and consistent chemical composition, resulting in a more compact and stronger structure. FA and MK-based geopolymers also appear to have a more consistent matrix or structure than FA-based geopolymers. Metakaolin’s chemical makeup aided in the dissolving of pozzolan, resulting in a more evenly dispersed structure.

### 3.3. Mineralogical Analysis

The XRD patterns for FA and MK-based geopolymers are shown in Figure 9. The largest peak, which can be observed at around 27°, was identified as quartz (SiO_2_). Amorphous phases are also indicated by multiple tiny peaks between 20° and 40°. Mullite, Maghemite, Haematite, and Aluminum Silicon Oxide (Al_2_ SiO_5_) are among the other phases discovered. Despite the increase in compressive strength seen in Figure 1, there is little variation between the three blends. On the other hand, the intensity of quartz in Mix 3 with a molar ratio of 2.5 is slightly higher. A rise in compressive strength usually corresponds to an increase in the intensity of crystalline products, particularly quartz. This is in line with the fact that Mix 3 has a higher compressive strength, as well as a denser matrix, as seen in Figure 1.

FA and MK-based geopolymers also appear to have a more consistent matrix or structure than FA-based geopolymers. Metakaolin’s chemical makeup aided in the dissolving of pozzolan, resulting in a more evenly dispersed structure. This was seen at a temperature of about 25°. Mullite, Maghemite, Haematite, and Aluminum Silicon Oxide (Al_2_SiO_5_) are among the other phases found. When XRD patterns of FA and MK-based geopolymers were compared to FA-based geopolymers (Figure 10), the peak/intensity for quartz in the FA and MK-based geopolymer was slightly greater. This is to be expected given the inequalities in compressive strength. Amorphous phases were also identified by minor uneven bumps between 20° and 43°. There is no discernible difference between the three mixes.

## 4. Conclusions

The following conclusions can be derived from the findings presented in this paper:The geopolymer made up of 70% fly ash and 30% metakaolin had a higher compressive strength than the geopolymer made up solely of fly ash.In comparison to geopolymers made entirely of fly ash, geopolymers made entirely of metakaolin appear to have a more homogeneous microstructure.When comparing geopolymer made up of 70% fly ash and 30% metakaolin to geopolymer made up of only fly ash, increasing the alkaline solution molar ratio resulted in a less porous structure with a lower number of unreacted fly ash and metakaolin particles.When compared to fly ash-based geopolymers, the geopolymer based on 70% fly ash and 30% metakaolin had a slightly higher peak/intensity for quartz.

## Figures and Tables

**Figure 1 materials-15-03732-f001:**
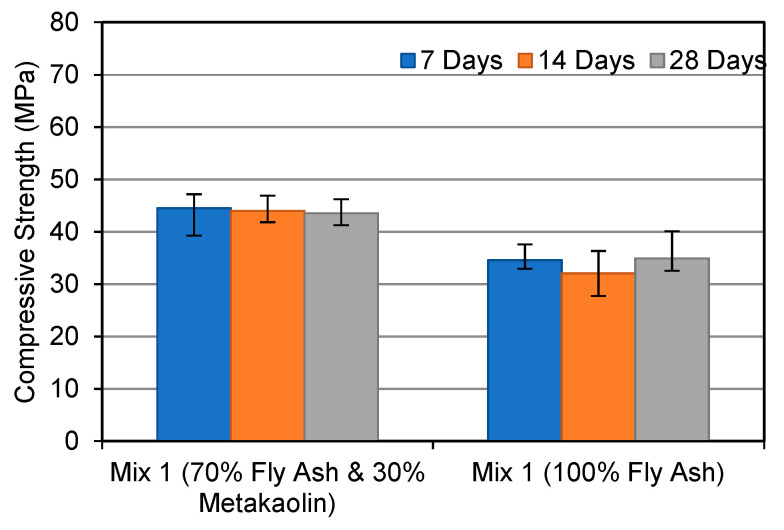
Compressive strength development at alkaline solution molar ratio 1.0.

**Figure 2 materials-15-03732-f002:**
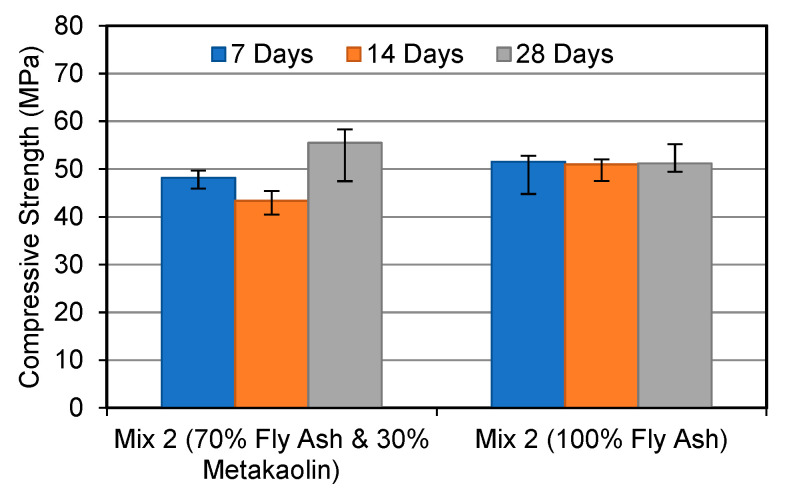
Compressive strength development at alkaline solution molar ratio 1.5.

**Figure 3 materials-15-03732-f003:**
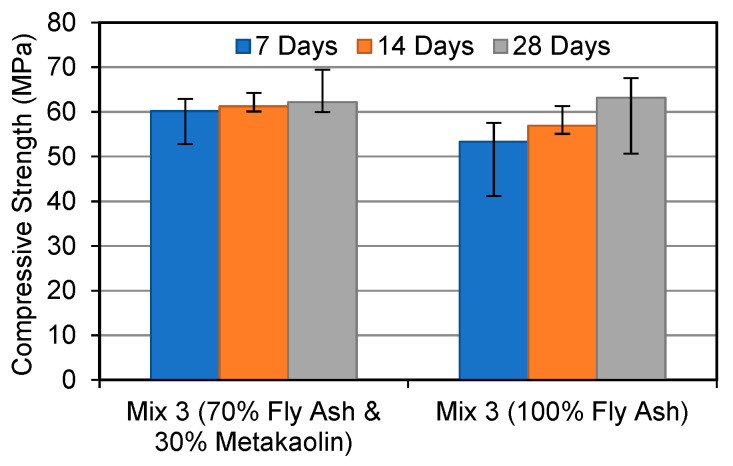
Compressive strength development at alkaline solution molar ratio 2.5.

**Figure 4 materials-15-03732-f004:**
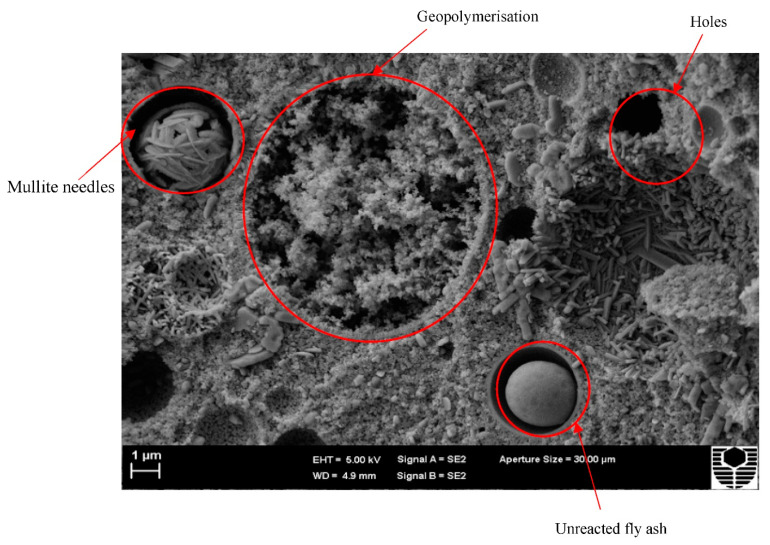
Microstructure of FA-based geopolymer.

**Figure 5 materials-15-03732-f005:**
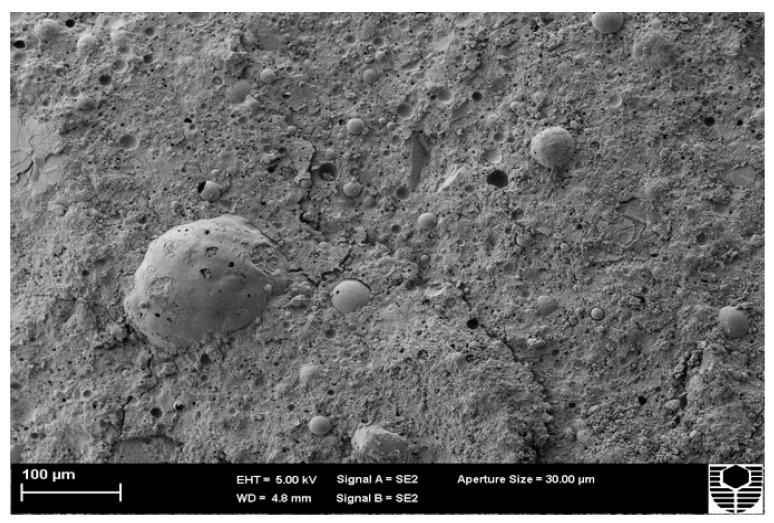
Microstructure of Mix 1 for FA and MK-based geopolymer.

**Figure 6 materials-15-03732-f006:**
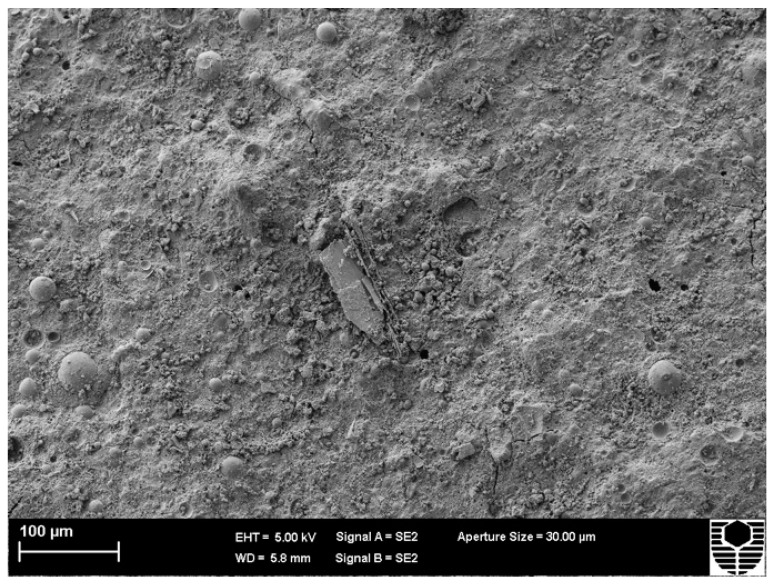
Microstructure of Mix 3 for FA and MK-based geopolymer.

**Figure 7 materials-15-03732-f007:**
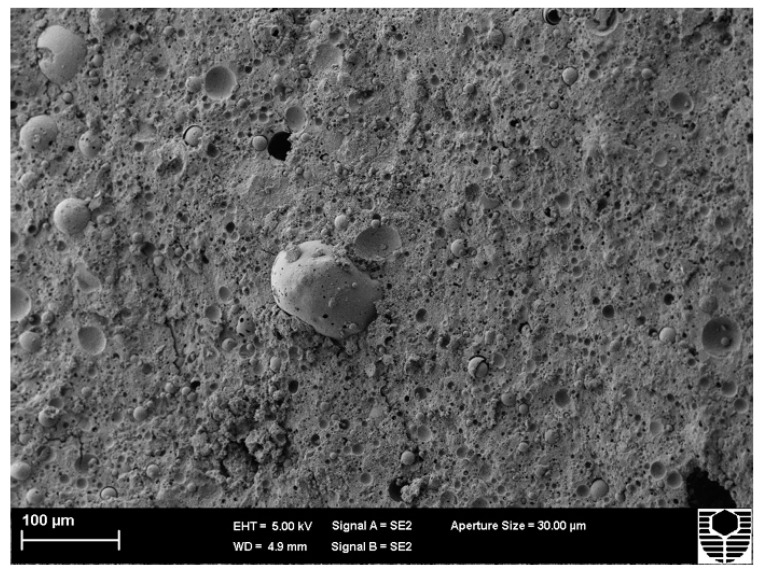
Microstructure of Mix 1 for FA-based geopolymer.

**Figure 8 materials-15-03732-f008:**
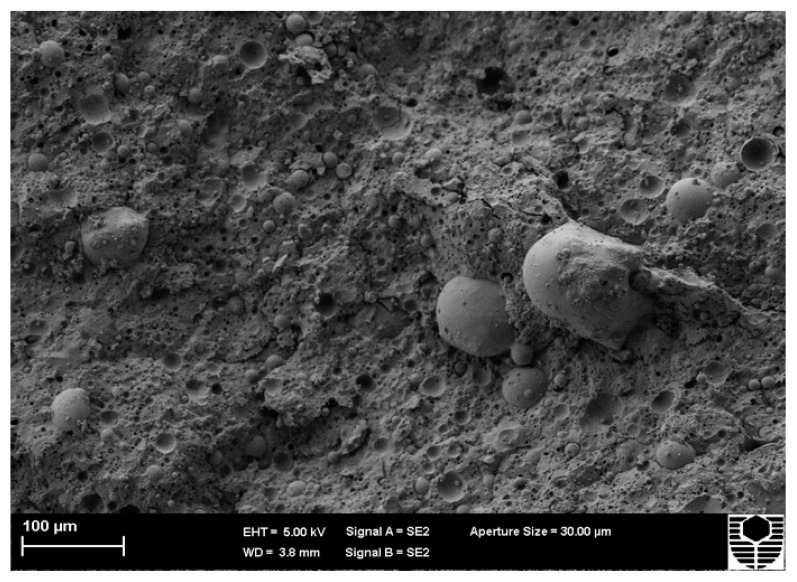
Microstructure of Mix 3 for FA-based geopolymer.

**Figure 9 materials-15-03732-f009:**
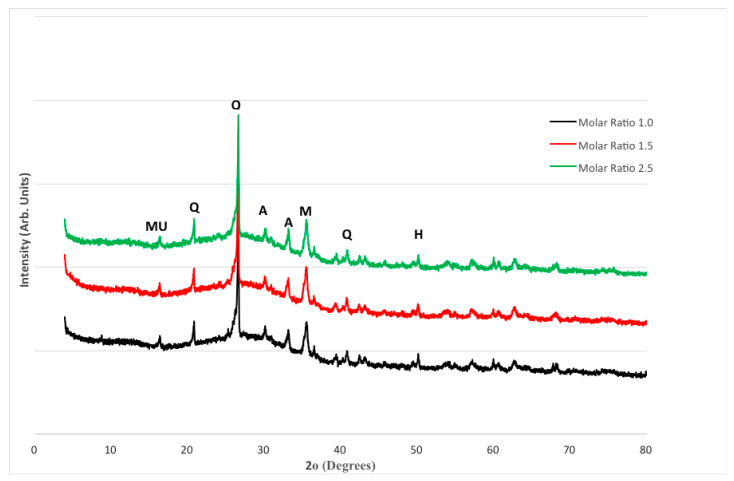
XRD Patterns for FA and MK-based geopolymers.

**Figure 10 materials-15-03732-f010:**
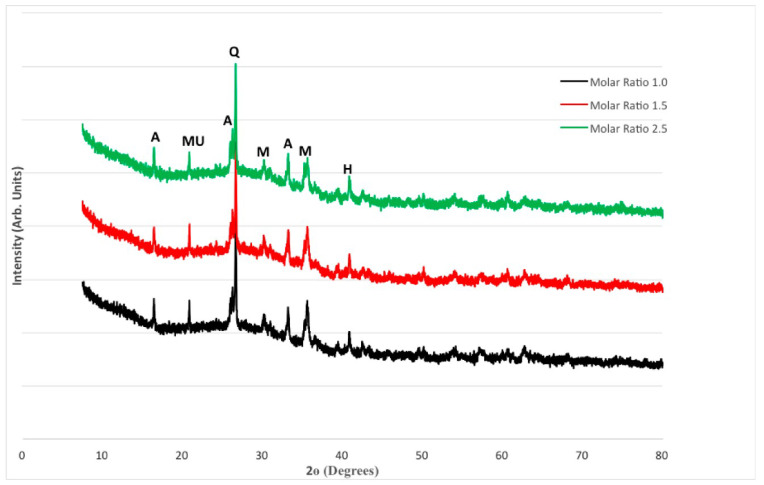
XRD Patterns for FA-based geopolymers.

**Table 1 materials-15-03732-t001:** Properties of fly ash and metakaolin.

Parameters	Fly Ash	Metakaolin
SiO_2_ (%)	51.80	52.10
Al_2_O_3_ (%)	26.40	41.00
Fe_2_O_3_ (%)	13.20	4.30
CaO	1.61	0.09
MgO	1.17	1.36
SO_3_	0.21	-
Na_2_O	0.31	0.01
K_2_O	0.68	0.62
P_2_O_5_	1.39	-
Loss of ignition (%)	0.50	0.50
Specific gravity	2.60	2.63

## Data Availability

Not applicable.

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
