# Peer review of "Strength and Microstructure of Geopolymer Based on Fly Ash and Metakaolin"

_materials, 2022, doi:10.3390/ma15103732_

Round 1
Reviewer 1 Report
A citation (product information) is required for the test instruments used in Section 2.2.
Line 77 misses '...days' at the end of the sentence.
Similarly, line 80 ends missing unit at the end "... of 80 5-80...?"
Figures 1 to 3 can be combined as one figure.
The labels of Figures 4 to 8 can be extended to explain what is being shown in the pictures.
Author Response
A citation (product information) is required for the test instruments used in Section 2.2.
Response: The product information is already there in Section 2.2
Line 77 misses '...days' at the end of the sentence.
Response: corrections made.
Similarly, line 80 ends missing unit at the end "... of 80 5-80...?"
Response: Corrections made.
Figures 1 to 3 can be combined as one figure.
Response: The figures separated for better discussion of the results.
The labels of Figures 4 to 8 can be extended to explain what is being shown in the pictures.
Response: Figurres 4 to 8 have been labelled.
Reviewer 2 Report
The paper "Strength and microstructure of Geopolymer based on fly ash and metakaolin" addresses a potential application of fly ash together with metakaolin in geopolymer materials. The topic fully adheres to the scope of publication of the materials, but in its current form it should not be considered for publication, I will cite some points that justify these issues:
(1) The general topic is widely known in the scientific community of the topic, the authors do not add any innovation, new methodology or necessary discussion to this study, this should not be considered in a prestigious journal. Note that this paper will hardly be cited, as there are countless other similar researches, just a simple database search;
(2) The experimental program is limited to mechanical strength testing in different molar ratios, and some microstructural characterizations, this is not an experimental program with the robustness necessary for publication, on the contrary, there are gaps that could be addressed regarding the use of other techniques, but the authors exploit something already accomplished;
(3) The abstract is simple, relevant information is lacking, the conclusion is incomplete and does not show advances in the literature, and the state of the art is highly limited, there are only 26 references.
Thus, based on the comments made, I do not see the possibility of accepting this manuscript.
Author Response
The revised paper has been improved.
Reviewer 3 Report
The paper presents a comparative experimental study on the performance of geopolymers produces with fly ash and those produced with a mixture of fly ash and metakaolin.
The paper is well-written, but a number of aspects should be addressed by the authors prior to eventual acceptance.
The paper framework and state of the art should be extended to include a number of more recent references. The manuscript currently contains 26 reference entries but only one is less than five years old.
Figures 1 to 3 illustrate results on the compressive strength assessment of the mixtures, to be compared with each other. For better reading of the graphs and comparison among each other, the vertical scale should be equal in all three figures.
Starting in page 3, lines 101 to 113, the authors discuss the results from Figures 1 to 3. This discussion should be reviewed, because at present the text is confusing and could be much improved. Please check this discussion and re-write it in a clearer way.
According to Figures 1 to 3, the strength has no tendency to increase with age, except maybe for Mix 3 100% fly ash. In the other series, the results at 7 days are similar to those at 28 days. Was this expected? Please comment on these results.
Figures 4 and 5 contain numbers (1) and (2) that are not cited in the text nor described in the Figure captions. Please check.
In conclusions, lines 180-181, the authors conclude that “geopolymer made up of 70% fly ash and 30% metakaolin had a higher compressive strength than the geopolymer made up of solely fly ash”. However, the results in Figures 1 to 3 suggest that this statement is valid only for Mix 1. When increasing the alkaline solution molar ratio, results for both geopolymer types are quite similar, when taking into account the dispersion of values registered. This dispersion is not commented in the paper, and it should be. Please check the conclusion.
In conclusions, lines 182-183, the authors conclude on results concerning “geopolymers made entirely of metakaolin”. However, no tests on samples with this characteristics were carried out or presented in the paper. Please check the conclusion.
Author Response
More recent references have been added in the revised paper.
Figures 1 to 3: The vertical scale made equal in all three figures.
Page 3, lines 101 to 113: Re-written.
Figures 4 and 5 contain numbers (1) and (2): Removed.
Conclusions: Checked.
Reviewer 4 Report
This paper looks more like a preliminary study than a research. The contribution of the paper in the area is not highlighted. The materials are insufficiently described; fly ash and metakaolin are not described at all in the Materials section. Nowadays, it is not enough just to test compressive strength….I believe you have much more results than it is shown here - this is your chance to complement the paper. A lot of papers covering geopolymers have been published recently — please add recent papers to the references.
Author Response
The description of fly ash and metakaolin has been added.
More recent references added.
Round 2
Reviewer 2 Report
ok.
Reviewer 3 Report
The authors addressed the reviewer recommendations and carried out the suggested changes in the manuscript.
Reviewer 4 Report
I still think that you have a small number of method applied and a small number of test results but you improved interpretation of your results....